# Molecular Detection of Novel *Borrelia* Species, *Candidatus Borrelia javanense*, in *Amblyomma javanense* Ticks from Pangolins

**DOI:** 10.3390/pathogens10060728

**Published:** 2021-06-09

**Authors:** Bao-Gui Jiang, Ai-Qiong Wu, Jia-Fu Jiang, Ting-Ting Yuan, Qiang Xu, Chen-Long Lv, Jin-Jin Chen, Yi Sun, Li-Qun Fang, Xiang-Dong Ruan, Teng-Cheng Que

**Affiliations:** 1State Key Laboratory of Pathogen and Biosecurity, Beijing Institute of Microbiology and Epidemiology, Beijing 100071, China; jiangbaogui@hotmail.com (B.-G.J.); jiangjf2008@gmail.com (J.-F.J.); 15605533218@163.com (T.-T.Y.); xuqiang09@nudt.edu.cn (Q.X.); tjjys2012@163.com (C.-L.L.); m18895632365@163.com (J.-J.C.); sunyi73@gmail.com (Y.S.); fang_lq@163.com (L.-Q.F.); 2Guangxi Zhuang Autonomous Region Terrestrial Wildlife Medical-aid and Monitoring Epidemic Diseases Research Center, Nanning 530028, China; wuaiqiong88@163.com; 3Academy of Forest Inventory and Planning, State Forestry and Grassland Administration, Beijing 100714, China

**Keywords:** emerging tickborne pathogens, *Borrelia*, *Amblyomma javanense*, ticks, *Manis javanica*, pangolins

## Abstract

A novel *Borrelia* species, *Candidatus Borrelia javanense*, was found in ectoparasite ticks, *Amblyomma javanense*, from *Manis javanica* pangolins seized in anti-smuggling operations in southern China. Overall, 12 tick samples in 227 (overall prevalence 5.3%) were positive for *Candidatus B. javanense*, 9 (5.1%) in 176 males, and 3 (5.9%) in 51 females. The phylogenetic analysis, based on the 16S rRNA gene and the flagellin gene sequences of the *Borrelia* sp., exhibited strong evidence that *Candidatus B. javanense* did not belong to the Lyme disease *Borrelia* group and the relapsing fever *Borrelia* group but another lineage of *Borrelia*. The discovery of the novel *Borrelia* species suggests that *A. javanense* may be the transmit vector, and the *M. javanica* pangolins should be considered a possible origin reservoir in the natural circulation of these new pathogens. To our knowledge, this is the first identification of a novel *Borrelia* species agent in *A. javanense* from pangolins. Whether the novel agent is pathogenic to humans is unknown and needs further research.

## 1. Introduction

*Borrelia*, a bacteria genus of the spirochete phylum, are the causative agents of major vector-borne diseases, including Lyme disease (LD) and relapsing fevers (RFs), which are transmitted by ticks and lice worldwide [1]. The LD and RF *Borrelia* species from two separate sister groups are considered two sister genera [2,3,4,5]. However, recent surveys have uncovered new *Borrelia* species and strains that do not belong to the LD or RF group. They form the third group, encompassing only two designated species, *B. turcica* and “*Candidatus Borrelia tachyglossi*”, and a few strains not taxonomically described [6,7,8,9,10,11,12,13,14,15,16,17]. The third *Borrelia* group has been detected in *Amblyomma*, *Hyalomma*, *Bothriocroton*, and *Ixodes* genera hard ticks, mainly associated with reptiles and echidna. However, only two genomes sequenced confirmed that these *Borrelia* are substantially different from the LD and RF *Borrelia* groups [15].

Here, we report a novel *Borrelia* species in *Amblyomma javanense* ticks collected from 25 *M. javanica* pangolins from uncertain countries in Southeast Asia, seized in anti-smuggling operations by Guangxi Customs officers in Fang Cheng Gang and Nan Ning cities, southern China, during the period October 2016–January 2018. We obtained the complete 16S rRNA gene (*rrs*) and the partial flagellin gene (*flaB*) by PCR and sequencing. All positive samples have 100% similar nucleotide sequences to each other. Both the *rrs* and the *flaB* gene sequences did not fall into either the Lyme disease *Borrelia* group or the relapsing fever *Borrelia* group but into another *Borrelia* group lineage in phylogenetic analysis.

## 2. Results

### 2.1. Identification of Borrelia sp. in Ticks

During the study period, we screened a total of 227 adult ticks captured from 25 *M. javanica* pangolins seized in anti-smuggling operations in southern China (Figure 1). The animals were brought to the wildlife medical-aid center during the veterinary examination, and the ticks were removed and preserved. All ticks were not engorged or not fully engorged and identified as *A. javanense* by morphologic features to the species level and the developmental stage by two entomologists (Y. Sun and R.-M. Xu). Twelve tick samples (overall prevalence 5.3%) were screened positive for *Borrelia*, 9 (5.1%) in 176 males and 3 (5.9%) in 51 females, with no significant gender difference. All nucleotide sequences of the 353-bp amplicons were sequenced and identical to one another. The nucleotide sequences of nearly entire *rrs* (1532 bp) were obtained by nested PCR for segmented amplification, sequencing, and splice according to the overlap area from 5 tick samples; all of them were identical to one another (GenBank accession no. MW889882) and similar to *Borrelia crocidurae* (GenBank accession no. CP004267), with 25 base-pair differences (1507/1532, homologous 98%). We also amplified and sequenced the partial flagellin gene (*flaB*) (GenBank accession no. MW916611) by nested PCR from 3 tick samples; these sequences were 100% homologous to one another and shown to be 92% (424/460) homologous with the corresponding sequence of the *Borrelia johnsonii* strain from bat tick *Carios kelleyi* from the US [18]. Both the *rrs* and *flaB* showed relatively large differences with the *Borrelia* sp. in GenBank and cannot be clustered into their *Borrelia* group. Therefore, they should be considered a separate branch or another *Borrelia* group.

### 2.2. Genetic Characteristics of Borrelia sp.

The *rrs* and *flaB* phylogenetic trees were constructed using the neighbor-joining method based on the trimmed *rrs* and *flaB* sequences generated in this study, together with 36 *rrs* sequences and 31 *flaB* sequences from other *Borrelia* spp. retrieved from GenBank. The phylogenetic trees based on nearly complete *rrs* (1401 bp; Figure 2) and partial *flaB* (466 bp; Figure 3) demonstrated that the *Borrelia* sp. identified from the *A. javanense* ticks in this study were clustered in a separate clade that was distinct from both LD *Borrelia* and RF *Borrelia* groups.

## 3. Discussion

Ticks (Acari: Ixodidae), which are obligate blood-feeding arthropods, are distributed worldwide from tropic to subarctic regions, capable of transmitting the broadest spectrum of pathogens, including bacteria, protozoa, fungi, nematodes, and viruses, to humans, livestock, and wildlife. Here, we report a non-described *Borrelia* species in *A. javanense* ticks attached below the scales of *M. javanica* pangolins distributed in south Asia, southeast Asia, and southern China. This novel *Borrelia* species is prevalent in this tick species, abundantly attached to the Malayan pangolins. The novel *Borrelia* species is divergent not only from the LD species and the RF species but also from a third *Borrelia* group, *B. turcica* and *B. tachyglossi*, the reptile- and echidna-associated *Borrelia* group that was recently described [19]. Sequence analysis showed that this novel *Borrelia* sp. perhaps has a unique genome and indicates that this novel *Borrelia* is an intermediate taxon between the LD and RF species. The designation “*Candidatus Borrelia javanense*” is proposed for this species.

We reported for the first time the identification of novel *Borrelia* agents in *A. javanense* from pangolins. *A. javanense* is an important ectoparasite of *M. javanica* pangolin. *A. javanense* has been found in almost all the pangolin species and could be the coendangered tick species on pangolins in Asia [20]. It is reported that the tick also infests various wild animals such as wild boar, lizards, python, skink, hill turtle, bat, hyena, bear, and sambar deer [21,22]. A previous study also reported that *A. javanense* could infect humans [23]. The geographical distribution of the tick is also very extensive. Voltzit and Keirans have reported *A. javanense* from Pakistan, India, Sri Lanka, Myanmar, Thailand, Vietnam, Malaysia, Singapore, Indonesia, Philippines, and China [24,25]. Among the 227 ticks tested in this study, 12 (overall prevalence 5.3%) were infected with the new agents, indicating that *A. javanense* may be acting as the tick vector of the newly identified *Borrelia* sp. There is a low probability of transmission to people, but whether it is pathogenic to humans is unknown and needs further exploration.

In conclusion, we have identified a novel *Borrelia* species, Candidatus *Borrelia javanense*, in ectoparasite ticks, *Amblyomma javanense*, from *M. javanica* pangolin. This is the first report of *Borrelia* agents in *A. javanense* from pangolins. Further studies should be conducted to isolate this bacterium and investigate its epidemiologic, genetic, and pathogenic features.

## 4. Materials and Methods

### 4.1. Ethics Statement

The Malayan pangolins studied here were rescued and treated by the Guangxi Zhuang Autonomous Region Terrestrial Wildlife Medical-Aid and Monitoring Epidemic Diseases Research Center under ethics approval (Wild Animal Treatment Regulation No. [2011] 85). The tick samples were collected following the procedure guideline (Pangolins Rescue Procedure, November 2016).

### 4.2. Tick Sample Collection and Identification

Ticks were collected from under the scales of the *M. javanica* pangolins seized by Guangxi Customs officers in anti-smuggling operations in Fang Cheng Gang city (108.35342° S, 21.76913° E) and Nan Ning city (108.33639° S, 22.87786° E). All the pangolins were from southeast Asian countries. Briefly, smugglers in the Sino-Vietnamese border collected the pangolins from Vietnam, Myanmar, Laos, Cambodia, Philippines, Malaysia, Indonesia, and Thailand and illegally introduced them into China. In the process of smuggling, the pangolins were intercepted and seized. During the rescue of the pangolins, all ticks, including engorged and not-engorged adults and nymphs, were removed from the body surface, preserved in a moist breathable bottle at 4 degrees, and sent to the Beijing Institute of Microbiology and Epidemiology for species identification and molecular analyses. All ticks were identified by entomologists Yi Sun and Rong-Man Xu. All identified ticks corresponded to *Amblyomma javanense*. For this study, a total of 227 adult ticks, including 176 males and 51 females, were collected, and 156 ticks that were not fully engorged were used for DNA isolation and molecular detection. Nymphs were not processed.

### 4.3. DNA Extraction, Borrelia-Specific PCR, and Sequencing

Ticks were surface-sterilized with 10% sodium hypochlorite and washed with sterile and DNA-free water and 70% ethanol. The ticks were grounded individually in 200 μL phosphate-buffered saline buffer, and total DNA was extracted with the DNeasy Tissue Kit (QIAGEN, Germantown, MD, USA). The ticks were subjected to two *Borrelia* genus-specific PCR assays (ABI9700, ). The *Borrelia*-specific nested-PCR assays were conducted by targeting the 16S rRNA gene (*rrs*) and the flagellin gene (*flaB*) (Table 1). Each 25 μL PCR reaction contained 1× PerfectTaq buffer, 2.5 mM MgCl2, 1 mM dNTPs, 400 nM of each primer, 1.25 U PerfectTaq polymerase, and 2 μL undiluted DNA. Both the primary and nested *Borrelia* 16S PCR assays were performed with the following thermal conditions: initial denaturation at 95 °C for 5 min, 35 cycles of denaturation at 95 °C for 30 s, annealing at 55 °C for 30 s, and extension at 72 °C for 1.5 min, and a final extension at 72 °C for 7 min. The flaB PCR assays were performed with an initial denaturation at 95 °C for 5 min, 35 cycles of denaturation at 95 °C for 30 s, annealing at 55 °C (primary and nested) for 30 s, and extension at 72 °C for 30 s, and a final extension at 72 °C for 7 min. Amplified PCR products were electrophoresed through 1.5% agarose gels, stained with ethidium bromide, and visualized under UV light. To avoid the risk of contamination, the DNA extraction, PCR reagent setup, amplification, and agarose gel electrophoresis were performed in separate rooms, and a negative control (distilled water) was concurrently included in each amplification. All the PCR products were purified with the QIAmp Gel Extraction Kit (QIAGEN, Germantown, MD, USA) and then directly sequenced on an automated DNA sequencer (3730 DNA Sequencer; Applied Biosystems, Carlsbad, CA, USA). We compared the sequences obtained with previously published sequences deposited in GenBank by using BLAST (http://blast.ncbi.nim.nih.gov/Blast.cgi, accessed on 15 June 2020).

### 4.4. Sequence Analysis

Phylogenetic analysis based on the 16S rRNA gene (*rrs*) and the flagellin gene *(flaB*) was conducted using MEGA7 software. The phylogenetic tree was constructed using the neighbor-joining method based on the trimmed sequences generated in this study, together with sequences from other *Borrelia* spp. retrieved from GenBank. The stability of the tree was evaluated by bootstrap analysis with 1000 replications. The 16S rRNA and flagellin nucleotide sequences determined in this study have been submitted to Genbank under accession numbers MW889882 and MW916611.

## Figures and Tables

**Figure 1 pathogens-10-00728-f001:**
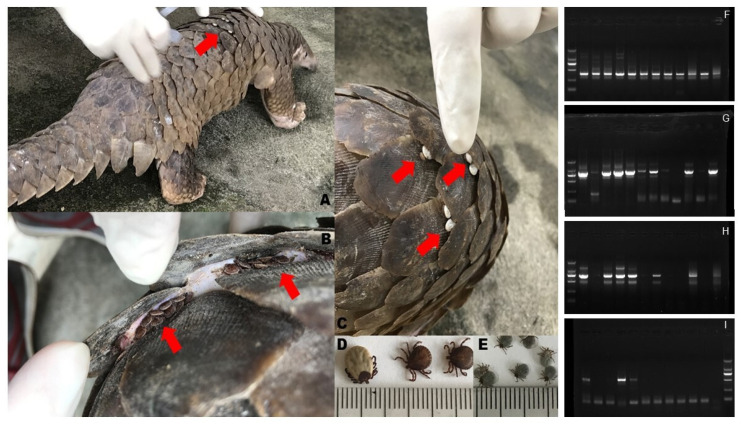
Illustrations of the host *M. javanica* pangolin, the attached ticks, *A. javanense*, below the scales, the removed female and male adult ticks, the nymph ticks (**A**–**E**), the electrophoretogram of *Borrelia* sp. screening for the 353-bp *rrs* (**F**), the 790-bp front half segment of full-length *rrs* (**G**), the 743-bp back half segment of full-length *rrs* (**H**), and the 506-bp *flaB* (**I**). The size of the DNA marker was 2000, 1000, 750, 500, 250, and 100 bp from top to bottom, respectively.

**Figure 2 pathogens-10-00728-f002:**
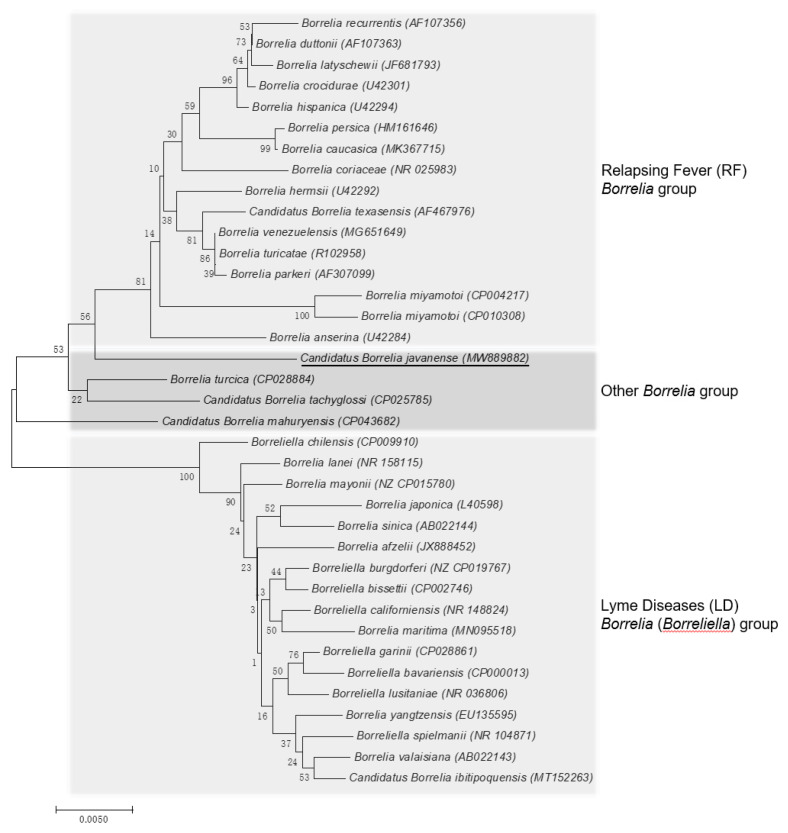
Phylogenetic tree based on nucleotide sequences of the 16S rRNA (1401-bp) genes of *Candidatus Borrelia javanense* in ectoparasite ticks, *A. javanense*, from *M. javanica* pangolins and the comparison sequences. Underline indicates the *Candidatus B. javanense* identified in this study; GenBank accession numbers are provided for all isolates. The neighbor-joining method was used to construct the evolutionary tree. The percentage of replicate evolutionary trees in which the associated taxa clustered together in the bootstrap test (1000 replicates) are shown next to the branches. The tree is drawn to scale, with branch lengths in the same units as those of the evolutionary distances used to infer the phylogenetic tree. The evolutionary distances were computed using the maximum composite likelihood method and are in the units of the number of base substitutions per site (MEGA7, 2015).

**Figure 3 pathogens-10-00728-f003:**
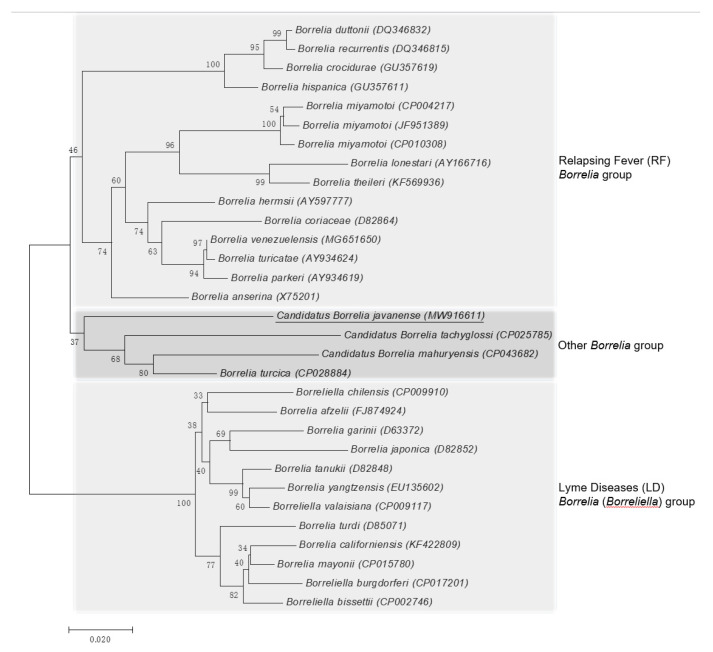
Phylogenetic tree based on nucleotide sequences of the flagellin gene (466-bp) of *Candidatus Borrelia javanense* in ectoparasite ticks, *A. javanense*, from *M. javanica* pangolins and the comparison sequences. Underline indicates the *Candidatus B. javanense* identified in this study; GenBank accession numbers are provided for all isolates. The neighbor-joining method was used to construct the evolutionary tree. The percentage of replicate evolutionary trees in which the associated taxa clustered together in the bootstrap test (1000 replicates) are shown next to the branches. The tree is drawn to scale, with branch lengths in the same units as those of the evolutionary distances used to infer the phylogenetic tree. The evolutionary distances were computed using the maximum composite likelihood method and are in the units of the number of base substitutions per site (MEGA7, 2015).

**Table 1 pathogens-10-00728-t001:** Primers used for *Borrelia*-specific 16S rRNA and flaB gene amplification in this study, including primer sequences, annealing temperature, and expected product size.

Gene	Primer	Sequence (5′-3′)	Annealing Temperature	Expected Product Size	Reference
16S rRNA	External				
	Bor-0F	AAAATAACGAAGAGTTTGATCCTGG			This study
	Bor-1533R	GTGATCCAGCCACACTTTCCAGTA	55 °C	1533 bp	This study
	Internal				
	Brm1	CGCTGTAAACGATGCACACTTGGTGTTAATC			[26]
	Brm2	CGGCAGTCTCGTCTGAGTCCCCATCT	60 °C	353 bp	[26]
	Brm1-R	GATTAACACCAAGTGTGCATCGTTTACAGCG	55 °C	790 bp/Bor-0F	This study
*flaB*	External				
	FLA120F	AGAATTAATMGHGCWTCTGATGATG			[26]
	Flab764R	GCATCTTCGATCTTTGAAAGTGACATATT	55 °C	645 bp	This study
	Internal				
	Flab625R	CTGGAGCTGCTTGAGCACCTTCT	55 °C	506 bp/FLA120F	This study

## Data Availability

The data presented in this study are openly available in MDPI.

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
