# Peer review of "Molecular Detection of Novel Borrelia Species, Candidatus Borrelia javanense, in Amblyomma javanense Ticks from Pangolins"

_pathogens, 2021, doi:10.3390/pathogens10060728_

Round 1

Reviewer 1 Report

Strengths: This study reports important new information. There is sparse information about Borrelia in Asia and this is an important contribution. The sequence analysis and techniques used are appropriate and nicely reported.

Weaknesses: There appears to be some confusion about the role of reservoirs and vectors.  Ticks can serve in both roles, but they are not properly defined in the paper.

It should be stated whether the ticks were engorged or unengorged at time of collection. If engorged, the test results may be detecting pathogens present in the pangolins’ blood. This could still be possible even if ticks were not fully engorged.

Was a blood sample collected from pangolins? If so, this should have been tested and reported.

The location is not listed from where the pangolins were smuggled in from.  They were obviously not from Fang Cheng Gang of Nan Ning city.  The location or country should be reported if known.

Line 162: Why were positive controls not used in the PCR? I understand the novel species was unknown, but a genera specific Borrelia or other species could have been used.

Line 69: List the genus and species of the bat tick from the United States

Line 125: What does “reservoir vector” mean? These terms should be better explained. It should also be mentioned the ticks may have tested positive because the pangolins were infected. Until vector transmission studies are conducted it will be unclear if Amblyomma javanense are competent vectors.

Minor grammar recommendations:

Line 19: belong – belongs

Line 20: alone Borrelia group – another Borrelia group

Line 34: and considered – are considered

Line 55: medical-aid center, during the – medical-aid center, and during

Line 64: and closed to Borrelia crocidurae - and similar to Borrelia crocidurae

Line 83: Figure – Figure 2

Line 85: clustered in an alone clade – clustered in a separate clade

Line 109: from the third Borrelia group -from a third Borrelia group

Line 119: phyton – python

Line 130: first reporting the Borrelia agents – first report of Borrelia agents

Line 145: (capitalize all parts of title) Beijing Institute of Microbiology and Epidemiology

Author Response

Point 1: Weaknesses: There appears to be some confusion about the role of reservoirs and vectors.  Ticks can serve in both roles, but they are not properly defined in the paper.

Response 1: We have rewritten the sentence in the revised manuscript, “A. javanense maybe the transmit vector and pangolins should be considered as possible origin reservoir in the natural circulation of the new pathogens.” (Line 21-22).

 Point 2: It should be stated whether the ticks were engorged or unengorged at time of collection. If engorged, the test results may be detecting pathogens present in the pangolins’ blood. This could still be possible even if ticks were not fully engorged. Was a blood sample collected from pangolins? If so, this should have been tested and reported.

Response 2: For the 227 ticks, some ticks were not engorged, some were not fully engorged. We have rewritten the sentence in the revised manuscript. (Line 56-57).In the study design, we have ever considered testing the blood of pangolins, but because the pangolins are endangered and are national first-level protected animal, we followed the Endangered Animals ethics and did not take blood or organ tests from it. But I think the blood test is important, and I look forward to making up for this test in the future

 Point 3: The location is not listed from where the pangolins were smuggled in from. They were obviously not from Fang Cheng Gang of Nan Ning city. The location or country should be reported if known.

Response 3: The reviewer's comment is right. All the pangolins were not from China, in fact, they are smuggled from some uncertain country in Southeast Asia, maybe Vietnam, maybe Myanmar. We have rewritten the sentence in the revised manuscript. (Line 43-44).

Point 4: Line 162: Why were positive controls not used in the PCR? I understand the novel species was unknown, but a genera specific Borrelia or other species could have been used.

Response 4: In order to prevent possible false positives, we did not add positive control during the PCR test. We have rewritten the sentence in the revised manuscript. (Line 162).

 Point 5: Line 69: List the genus and species of the bat tick from the United States

Response 5: As suggested, we have filled in the information of tick species in the revised manuscript. (Line 70). Candidatus Borrelia johnsonii is a species previously detected only in the bat tick, Carios kelleyi. The related reference is ‘Surveillance for and Discovery of Borrelia Species in US Patients Suspected of Tickborne Illness. Clin Infect Dis. 2018 Jun 1;66(12):1864-1871. doi: 10.1093/cid/cix1107.

 Point 6: Line 125: What does “reservoir vector” mean? These terms should be better explained. It should also be mentioned the ticks may have tested positive because the pangolins were infected. Until vector transmission studies are conducted it will be unclear if Amblyomma javanense are competent vectors.

Response 6: Thanks for reviewer’s rigorous suggestion. Because the vector transmission studies were not conducted. Whether Amblyomma javanense is reservoir vector for the new pathogen is unknown. We have rewritten the sentence in the revised manuscript. (Line 21-22, 125)

Point 7: Line 19: belong – belongs

Response 7: As suggested, we have revised the word in the revised manuscript. (Line 19).

 Point 8: Line 20: alone Borrelia group – another Borrelia group

Response 8: As suggested, we have revised the word in the revised manuscript. (Line 20).

Point 9: Line 34: and considered – are considered

Response 9: As suggested, we have revised the word in the revised manuscript. (Line 34)

Point 10: Line 55: medical-aid center, during the – medical-aid center, and during

Response 10: As suggested, we have revised the word in the revised manuscript. (Line 34)

Point 11: Line 64: and closed to Borrelia crocidurae - and similar to Borrelia crocidurae

Response 11: As suggested, we have revised the word in the revised manuscript. (Line 65).

Point 12: Line 83: Figure – Figure 2

Response 12: As suggested, we have revised the word in the revised manuscript. (Line 84).

Point 13: Line 85: clustered in an alone clade – clustered in a separate clade

Response 13: As suggested, we have revised the word in the revised manuscript. (Line 85).

Point 14: Line 109: from the third Borrelia group -from a third Borrelia group

Response 14: As suggested, we have revised the word in the revised manuscript. (Line 109).

Point 15: Line 119: phyton – python

Response 15: As suggested, we have revised the word in the revised manuscript. (Line 119).

Point 16: Line 130: first reporting the Borrelia agents – first report of Borrelia agents

Response 16: As suggested, we have revised the word in the revised manuscript. (Line 130).

Point 17: Line 145: (capitalize all parts of title) Beijing Institute of Microbiology and Epidemiology

Response 17: As suggested, we have revised the word in the revised manuscript. (Line 145).

Reviewer 2 Report

The genospecies in Borrelia genus has been grouped into three different sub-groups includes 1) Lyme disease borreliae, 2) relapsing fever borreliae, and 3) the third group of borreliae. In this study, Gui et al. identified a new species of Borrelia that appears to be in the third group, supporting the above-mentioned grouping concept. The manuscript was well written and easy to read. The rationale to support this new species is also strong. I only have few comments that need to be clarified.

In Fig. 2 and 3, depending the location of the root on the phylogeny, the new species of Borrelia may look like in the RF Borrelia group (based on 16srRNA) or the third group (based on flaB). How did the author make the conclusion that this species is in the third group rather than in RF Borrelia group?

In addition, few things may need to be clarified for the phylogenetic tree reconstruction. For example, how much support for each node on the tree should be indicated (eg. posterior probability support). The rationale of the method (Neighbur-Joining method) that was chosen to generate the tree should also be mentioned. 

Author Response

Point 1: In Fig. 2 and 3, depending the location of the root on the phylogeny, the new species of Borrelia may look like in the RF Borrelia group (based on 16srRNA) or the third group (based on flaB). How did the author make the conclusion that this species is in the third group rather than in RF Borrelia group?

Response 1: Because the 16sRNA gene is a relatively conservative gene sequence. The evolutionary distance of the gene between the new Borrelia species and the other Borrelia sp. in the RF Borrelia group is ≥ 25 bases,similarity only 98% or less. Such a long evolutionary distance is not enough to represent a group in the current cluster analysis of RF Borrelia group. The exact taxonomic status of the new species maybe needs to be more study in the future, such as pathogen isolation, whole genome sequencing, or multisite genes cluster analysis.

Point 2: In addition, few things may need to be clarified for the phylogenetic tree reconstruction. For example, how much support for each node on the tree should be indicated (eg. posterior probability support). The rationale of the method (Neighbur-Joining method) that was chosen to generate the tree should also be mentioned.

Response 2: As suggested, we have replaced the phylogenetic tree, each node on the tree was indicated. (Figure 2 and 3). Neighbur-Joining method was used and the evolutionary distances were computed using the Maximum Composite Likelihood method. The figure 2 and 3 legends were rewritten in the revised manuscript.